# Ligand-Assisted Formation of Graphene/Quantum Dot Monolayers with Improved Morphological and Electrical Properties

**DOI:** 10.3390/nano10040723

**Published:** 2020-04-11

**Authors:** Aleksandr P. Litvin, Anton A. Babaev, Peter S. Parfenov, Aliaksei Dubavik, Sergei A. Cherevkov, Mikhail A. Baranov, Kirill V. Bogdanov, Ivan A. Reznik, Pavel O. Ilin, Xiaoyu Zhang, Finn Purcell-Milton, Yurii K. Gun’ko, Anatoly V. Fedorov, Alexander V. Baranov

**Affiliations:** 1Center of Information Optical Technology, ITMO University, St. Petersburg 197101, Russia; a.a.babaev@ifmo.ru (A.A.B.); psparfenov@itmo.ru (P.S.P.); adubavik@itmo.ru (A.D.); s.cherevkov@itmo.ru (S.A.C.); mbaranov@itmo.ru (M.A.B.); bogdanov.k@niuitmo.ru (K.V.B.); ivanreznik1993@itmo.ru (I.A.R.); 182845@niuitmo.ru (P.O.I.); a_v_fedorov@mail.ifmo.ru (A.V.F.); a_v_baranov@mail.ifmo.ru (A.V.B.); 2College of Materials Science and Engineering, Jilin University, Changchun 130012, China; zhangxiaoyu@jlu.edu.cn; 3School of Chemistry and CRANN Trinity College Dublin, Dublin 2, Dublin D02 PN40, Ireland; fpurcell@tcd.ie (F.P.-M.); IGOUNKO@tcd.ie (Y.K.G.)

**Keywords:** quantum dots, graphene, self-assembly, spectroscopy, monolayers, charge transport

## Abstract

Hybrid nanomaterials based on graphene and PbS quantum dots (QDs) have demonstrated promising applications in optoelectronics. However, the formation of high-quality large-area hybrid films remains technologically challenging. Here, we demonstrate that ligand-assisted self-organization of covalently bonded PbS QDs and reduced graphene oxide (rGO) can be utilized for the formation of highly uniform monolayers. After the post-deposition ligand exchange, these films demonstrated high conductivity and photoresponse. The obtained films demonstrate a remarkable improvement in morphology and charge transport compared to those obtained by the spin-coating method. It is expected that these materials might find a range of applications in photovoltaics and optoelectronics.

## 1. Introduction

Graphene and its derivatives possess excellent conductivity, optical transparency, flexibility and stability. These exceptional properties make them attractive for solar cell fabrication [1,2,3,4,5,6]. In particular, graphene oxide (GO) and reduced graphene oxide (rGO) can be used for hole- (HTL) and electron-transporting (ETL) layers in organic and hybrid solar cells [4,7]. The doping of rGO and GO allows a change in its conductivity and the tuning of the Fermi level, both of which facilitate charge transfer [8]. Environmental safety and solution processability are additional advantages of graphene derivatives.

Graphene derivatives have been successfully used in the field of near-infrared (NIR) optoelectronics. Lead sulphide (PbS) quantum dots (QDs) have become an indispensable material for NIR optoelectronics because of their high absorption and processability [9,10,11]. The combination of graphene and PbS quantum dots (QDs) has already been successfully utilized in solar cells [12,13,14,15] and photodetectors [16,17,18,19,20]. The introduction of carbon allotropes into PbS-based devices can improve their performance and stability [14,21,22].

Beatriz Martín-García et al. have recently developed a method for attaching PbS QDs to the GO and rGO using short linking molecules [23]. (3-Mercaptopropyl) trimethoxysilane (MPTS) binds tightly to rGO/GO, while the thiol group remains free for attaching any metal atoms on the QD surface. 1,2-ethanedithiol (EDT)-treated layers of rGO–PbS, prepared in the same way, were used in heterojunction solar cells and brought superior device stability in the moisture atmosphere [14]. It has been recently shown that MPTS-functionalized rGO interlayers improve the morphological and electrical parameters of PbS-based solar cells. The functionalized rGO interlayers were introduced between tetrabutylammonium iodide-treated PbS layers and allowed them to achieve a superior film quality which resulted in a higher charge carrier concentration, lower trap state density, and the blocking of the minor charge carriers [24]. However, the spin-coating method leads to the waste of material and is problematic when upscaling the dimensions of a device. Moreover, in the case of graphene derivatives of micron sizes, it is a challenge to obtain high quality films. Therefore, the development of the new methods towards high quality PbS–graphene layers is an urgent task for further improvement of these devices.

The Langmuir–Blodgett (LB) technique was developed for the deposition of a compact monolayer of different molecules for electronics applications [25]. This method also allows the fabrication of close-packed thin QD films [26]. One of the main advantages of this method is precise control over the thickness of the films. In addition, the method ensures a dense packing of deposited substances and does not require unnecessary consumption of material.

In this work, we have used the Langmuir–Blodgett technique to fabricate thin films of PbS QDs, attached to rGO by the MPTS linker (LB films). Using this approach, a one-step fabrication of large-area smooth films can be easily fabricated, the properties of which have been readily tuned by post-deposition ligand exchange using a range of ligands. These rGO–PbS films have been compared to the more regularly utilized spin-coating method, with this novel approach producing films that demonstrate a remarkable improvement in morphology and charge transport.

## 2. Materials and Methods

### 2.1. Materials and Samples Preparation

Precursors PbO (99.999%), hexamethyldisilathiane (TMS), methylammonium iodide (MAI), mercaptopropionic acid (MPA), 1,2-ethanedithiol (EDT), tetrabutylammonium iodide (TBAI), dimethylformamide anhydrous (DMF) were all purchased from Sigma-Aldrich (Schnelldorf, Germany); acetonitrile was purchased from AppliChem (Darmstadt, Germany), octadecene (ODE) was obtained from Acros Organics (Fair Lawn, NJ, USA), oleic acid (OlAc) from Fisher Chemicals (Ward Hill, MA, USA), ethanol (EtOH), methanol (MeOH), and chloroform were purchased from Vekton (Saint Petersburg, Russia). Reduced graphene oxide (rGO) stabilized with poly(sodium 4-styrenesulfonate) was purchased from Sigma-Aldrich (Schnelldorf, Germany) as aqueous solution and used as received. A total of 95% (3-Mercaptopropyl) trimethoxysilane (MPTS) was purchased from Sigma-Aldrich (Schnelldorf, Germany).

PbS QDs were synthesized using the hot injection method described elsewhere [27]. To obtain 4.0 nm PbS QDs, 0.46 g of PbO was added to 2 g OlAc and 10 g ODE. The mixture obtained was degassed at 90 °C for 4 h. After flushing the set-up with argon, the temperature was raised to 105 °C to produce ~4.0 nm QDs. The S-precursor (210 μL TMS in 5 g ODE) was swiftly injected and stopped after 10 min. The purification of the QDs was fulfilled by the addition of thrice as much acetone, centrifuged for 10 min at 6000 rpm, and redispersed in toluene or hexane. The size of QDs was determined by absorption spectroscopy [28,29].

### 2.2. rGO Functionalization

The functionalization of rGO with the MPTS was provided using the methods reported in [23], Briefly, 12 mL of rGO dispersion (0.5 mg/mL) in EtOH was sonicated for 30 min and subsequently functionalized by reflux at 60 °C during 15 h with 1.5 mL of MPTS. The resulting dispersion was washed with EtOH to remove the excess MPTS and centrifuged at 6000 rpm for 3 min, and redispersed in EtOH by sonication (30 min).

### 2.3. PbS Linking

For the PbS-rGO hybrid preparation, the PbS QD solution in toluene was mixed with functionalized rGO solution in EtOH in a proportion of 1 mg of rGO to 0.8 nmol of QDs. The mixture was vortexed for 4 min to achieve effective mixing. Following this, the sample was cleaned using toluene and centrifugation at 6000 rpm for precipitation to remove unattached QDs and then redispersed in chloroform. 

### 2.4. Solution-Phase Ligand Exchange

For the MAI phase ligand exchange process, 1.5 mL of a 0.04 M MAI in 2:1 toluene–DMF solution was added at rate of (3 µL·s^−1^) to 4 mL of the PbS CQD solution with a concentration of 50 mg/mL, under gentle stirring. The resulting solution was kept under an inert atmosphere for 12 h to allow for the completion of the exchange. Following this, MeOH was used to precipitate the solution and, finally, the solution was redispersed in hexane [30]. The MPA-capped QDs were prepared by mixing oleic-capped QDs in hexane with 12.6 mM DMF MPA solution. After mixing, the resulting solution separated in two phases, and MPA-capped QDs were obtained from the low-polar phase and washed by centrifugation [31].

### 2.5. LB Film Deposition

Thin films were formed by the Langmuir–Blodgett method using the KN 2002 setup (KSV NIMA, Gothenburg, Sweden). Deionized water with a conductivity of ~18 MΩ/cm was used as a subphase. Before the formation of thin films; the surface of the subphase was cleaned of contamination with an air pump. The level of surface purity of the subphase was controlled by monitoring the surface pressure before thin film formation, which did not exceed 0.1 mN/m. The formation of the thin film began with the drop-casting of the CQD solution on the surface of the subphase, followed by 20 min of evaporation of the organic solvent [32]. After that, the formation of a thin film began by reducing the distance between the barriers, and, therefore, a decrease in the surface area occupied by the film. During this process, an increase in the surface pressure of the film occurs. When the desired surface tension is achieved, the film can be transferred to the substrate by lifting it with a speed of 1 mm/min. The process of film deposition is described in detail in Appendix A. Using this setup, the film was simultaneously transferred to six substrates with sizes of 3.0 × 2.0 cm^2^. The method is reproducible and scaled up, and the limitation comes from the geometrical size of the bath and sample holder.

### 2.6. Post-Deposition Ligand Exchange

TBAI in methanol (10 mg/mL) and EDT in acetonitrile (0.04 vol. %) were used for the post-deposition ligand exchange. The solutions were applied on the film for 30 s, then spin-coated at 2500 rmp and washed with pure acetonitrile twice [33].

### 2.7. Microscopy

A Merlin–Zeiss scanning electron microscope (SEM) was used to obtain the SEM images. A Solver Pro-M (NT-MDT, Moscow, Russia) atomic-force microscope (AFM) was used to study the morphology and thickness of thin films. To obtain optical images, a Leica optical microscope with backscattering geometry, 5× (0.12 numerical aperture) and 50× (0.75 numerical aperture) objective lenses were used.

### 2.8. Spectroscopy

A purpose-built setup for PL analysis in the NIR spectral region was used for characterization of PbS and rGO–PbS solutions [34]. A total of 5 mW of 530 nm LED was used in the steady-state mode, and 3 mW of 532 nm laser with a pulse duration of 1 ns was used in the transient mode. The decay curves were recorded using 1024 points with a step of 2–4 ns, so that the fastest estimated decay time was at least four times longer than the time resolution. A Tensor 27 (Bruker, Billerica, MA, USA) in attenuated total reflectance mode was used for the Fourier-transform infrared spectroscopy (FTIR) measurements.

### 2.9. J-V Measurements

Conductivity measurements were carried out using a Keithley 2636B source measure unit with the four-wire sensing method. To determine the conductivity of thin films, we measured the current as a function of applied voltage on films deposited on a pre-patterned indium-tin-oxide (ITO) glass substrate. We etched the 2 mm strip in the middle of the 12 × 12 mm^2^ ITO glass (8 Ω/sq sheet resistance) to make a channel between two ITO contacts. The clean substrate showed a leakage current less than 50 pA at 20 V voltage. Then these substrates were covered by the QD and rGO-QD films. To detect light sensitivity, we used a halogen lamp (color temperature 2800 K) with a light intensity in the visible range of 3 W/m^2^ [35].

## 3. Results and Discussion

To study the optical properties of PbS QDs and the rGO–PbS hybrid, we prepared the solutions of 4.0 nm oleic acid-capped PbS QDs (PbS-OA), QDs treated with MPTS, and rGO–PbS hybrids linked to each other with MPTS. The rGO–PbS hybrid solution was prepared with an rGO/PbS weight ratio of ~1/150.

Then, LB films from pure rGO and rGO–PbS solutions were prepared on an ITO-covered glass substrate. A drop of a solution of QDs and rGO, previously attached to each other, was placed at the liquid/air interface. The region is limited by sub-barriers, the approach of which increases the surface tension of the film. With a certain surface tension, a monolayer can form, which will be reflected in the dependence of tension on the film area (see Appendix A). Then, the formed monolayer is transferred to the substrate. The process of preparation of hybrid rGO–PbS thin films is shown in Figure 1. Pure rGO forms aggregates and cannot be used for LB film deposition. In contrast to that, the rGO–PbS hybrid solution allows the deposition of a large-area smooth film. Optical and SEM images of the hybrid rGO–PbS film obtained by the LB method (LB film) are shown in Figure 2a,b. Different colors on the optical images shown in Figure 2a demonstrate the different numbers of film layers. The SEM image presented in Figure 2b shows the area where the formation of the film starts.

To confirm the existence of both PbS QDs and rGO in the LB films, Raman and PL spectroscopies were used. Typical spectra from 4.0 nm PbS QDs in an LB film is shown in Appendix A. Transient PL measurements show that the PL lifetime of the QDs in an LB film drastically decreases in comparison with MPTS-treated QDs, indicating a strong charge transfer from QDs to rGO (Appendix A). It should be noted that MPTS-treated PbS QDs in solution show slightly longer PL decay times compared to PbS-OA. This can be attributed to the additional passivation of their surface through attaching the SH- group of MPTS to unpassivated lead atoms on the QD surface.

To estimate the thickness of the obtained LB rGO–PbS film, an atomic force microscopy (AFM) study was performed. AFM images and corresponding cross-sections are shown in Figure 3a,b. The smooth morphology of the film indicates the uniform distribution of QDs on the graphene surface. The obtained thickness of 13–22 nm indicates the formation of 1–2 monolayers of PbS QDs attached to rGO from both sides, which was also supported by SEM images shown in Figure 3c,d. The LB rGO–PbS films demonstrate a superior morphology compared to the rGO–PbS films prepared by the spin-coating method (Appendix A). The average roughness of the film decreases from 2.9 ± 1.0 nm to 1.7 ± 0.5 nm when the LB method is used.

To understand the mechanism underlying the film formation, we modified the procedure of hybrid solution preparation. First, we performed solution-phase ligand exchange with MAI and MPA to modify QDs properties. Such procedures are useful for optoelectronic device fabrication and allow the tuning of highest occupied (HOMO) and lowest unoccupied (LUMO) molecular orbitals energy positions, doping densities and charge mobilities [9,36]. After the ligand exchange, the QD solution was functionalized with MPTS rGO in the same manner. After precipitation and dissolution, the hybrid solutions were used for LB film fabrication. However, films with very low quality were obtained. In these two cases, the PbS QDs and graphene form aggregates, which produce cracks in the LB film, as can be seen from the SEM images shown in Appendix A. In the case of MPA-capped PbS QDs, we also expected that the bounding to the MPTS-rGO had a low efficiency, since the MPA is linked to the surface of the PbS QD using the thiol functional group as well. We can speculate that the removal of oleic acid during the ligand exchange plays a crucial role for the film formation. It has been recently shown that the amount of oleic acid can influence the self-organization of QD superstructures [37,38,39,40,41,42].

Indeed, the FTIR spectra demonstrate that oleic acid remains on the surface of the PbS QDs after rGO attachment. Figure 4 shows the FTIR spectra from pure oleic acid, PbS QDs, and the rGO–PbS hybrid. For the measurements of a hybrid material, a drop of concentrated solution was placed onto a diamond crystal, and the spectra were obtained after solution evaporation. Typical modes at 720, 1464, 2853, and 2923 cm^−1^ are related to oleic acid and can be observed in all cases. A typical O-H mode is observed for pure rGO and hybrid rGO–PbS samples as shown in Figure 5a. Then, FTIR measurements were performed for LB rGO–PbS film fabricated on a glass substrate. The traces of oleic acid can be observed in the peaks at 2853 and 2923 cm^−1^, which are shown in Figure 5b. This indicates that MPTS replaces the native ligands partly, and that some amount of oleic acid is present in the sample.

To confirm the critical role that oleic acid plays in the formation of high-quality films, we modified the rGO/PbS weight ratio from 1/150 to 1/15. The SEM images of the obtained LB film are shown in Appendix Ac. This sample consists of areas with high quality films and bare substrates, meaning that the lower concentrations of PbS QDs and therefore oleic acid, respectively, are only enough to enable the formation of island-like films.

However, insulating oleic acid ligands on the PbS QD surface for a long time prevents effective charge transfer between the QDs and must be removed during optoelectronic device fabrication. To perform post-deposition ligand exchange, the films were placed on a spin-coater, with TBAI and EDT treatments performed. The quality of the films remained perfect after the exchange procedure (Appendix A). As a reference, we made two films with a similar thickness from the rGO–PbS solutions, using the spin-coating technique, and performed a similar ligand exchange procedure. For conductivity measurements, a 2-mm stripe was etched on an ITO surface on this series of samples to perform the analysis. The current–voltage characteristics (I–V curves) are obtained for EDT- and TBAI-treated samples, prepared using the LB and spin-coating techniques, as shown in Figure 6.

It can be seen that the LB films demonstrate much higher currents under the same voltage. To compare the LB and spin-coated samples, we calculated the specific conductance of the films, which takes into account their geometrical parameters. The calculated values are listed in Table 1. It can be seen that the LB films possess superior parameters for device fabrication. Therefore, this result is interpreted to be due to the packing thickness, with the LB technique enabling the production of more densely packed QD-functionalized rGO sheets. Under white light illumination, an increase in current was observed, which indicates the effective photoelectrical response of the hybrid rGO–PbS films (Appendix A). This demonstrates that this approach can be used for the fabrication of efficient large-scale optoelectronic devices operating in the NIR spectral region.

## 4. Conclusions

In this study, we have shown a ligand-assisted formation of high-quality large-area hybrid rGO–PbS QD films with a thickness of one to two monolayers. We have demonstrated that the presence of oleic acid on the PbS QD surface plays a critical role in the film formation. Using this method, several substrates can be covered by rGO–PbS film simultaneously. In addition, the film formation and its properties can be easily tuned by standard procedures of post-deposition ligand exchange. The obtained films demonstrate superior morphological and electrical properties compared to the samples fabricated by the conventional spin-coating method. We believe that the proposed process can be used for the fabrication of hybrid monolayers of different types of QDs bonded to 2D materials for their use in optoelectronic devices.

## Figures and Tables

**Figure 1 nanomaterials-10-00723-f001:**
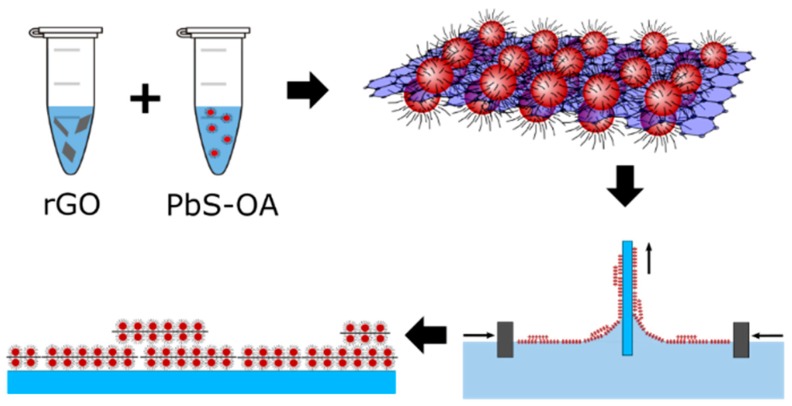
Schematic presentation of the preparation of thin rGO–PbS films.

**Figure 2 nanomaterials-10-00723-f002:**
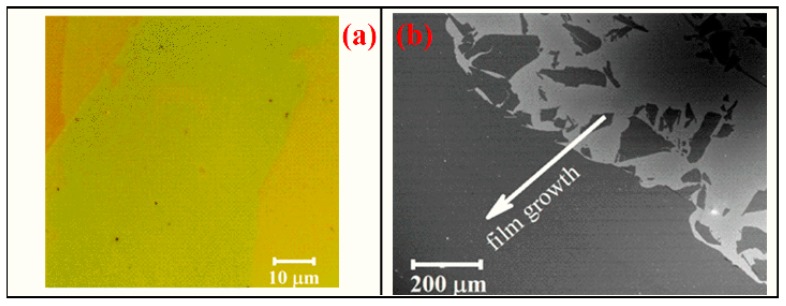
Optical (**a**) and SEM (**b**) images of the reduced graphene oxide (rGO)–PbS LB film.

**Figure 3 nanomaterials-10-00723-f003:**
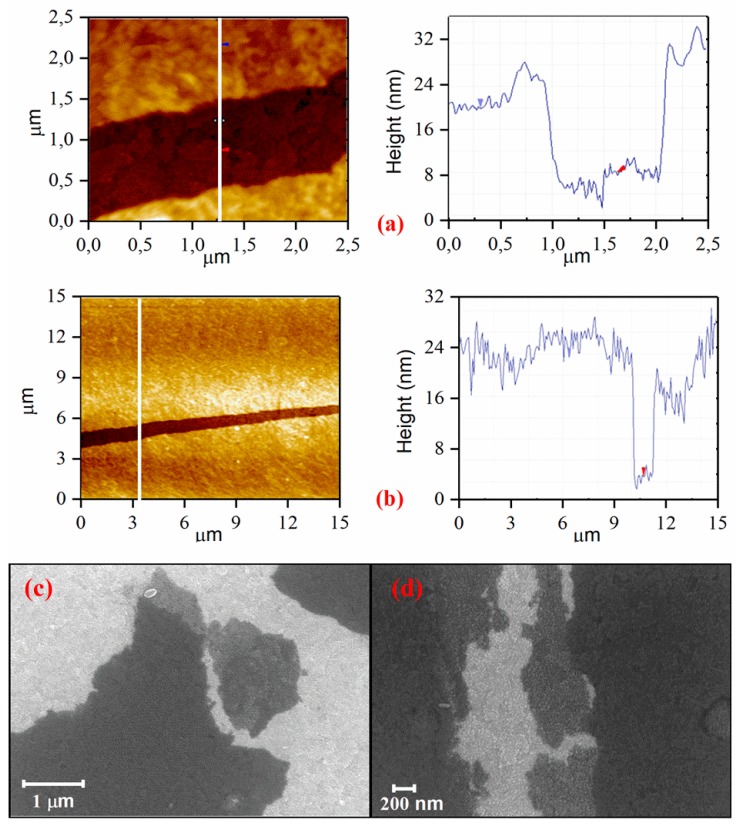
Atomic force microscopy (AFM) images and corresponding cross-sections (**a**,**b**), and SEM (**c**,**d**) images of hybrid rGO–PbS LB film.

**Figure 4 nanomaterials-10-00723-f004:**
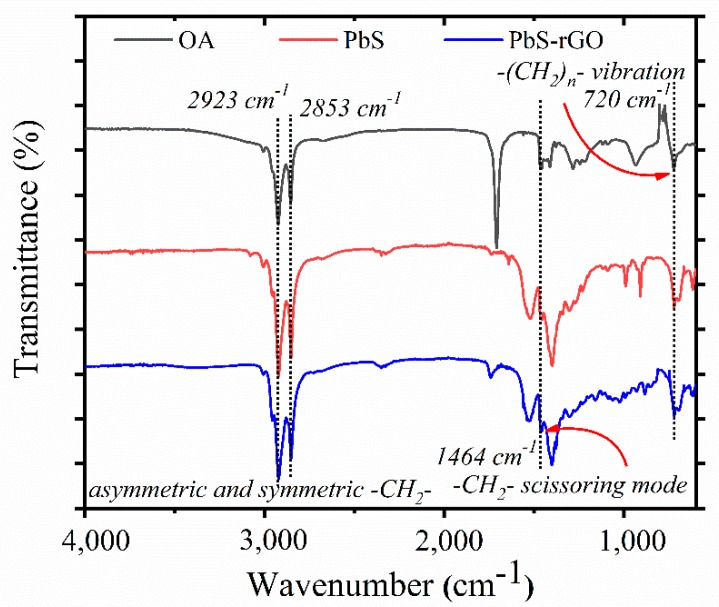
FTIR spectra of oleic acid, PbS QDs and rGO–PbS hybrid, indicating the presence of oleic acid.

**Figure 5 nanomaterials-10-00723-f005:**
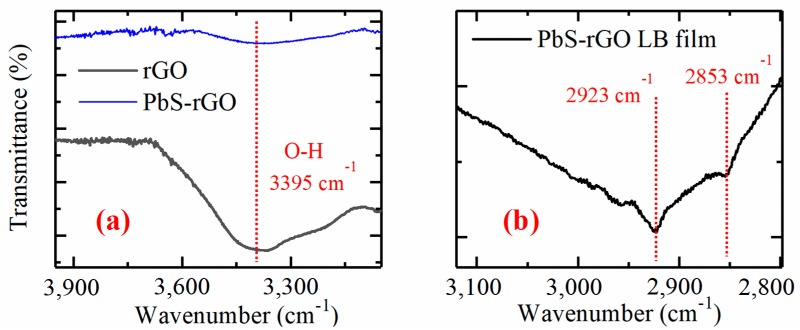
FTIR spectra of rGO–PbS hybrid indicating the O-H mode typical for rGO (**a**). FTIR spectra of rGO–PbS LB film indicating the traces of oleic acid on QD surface (**b**).

**Figure 6 nanomaterials-10-00723-f006:**
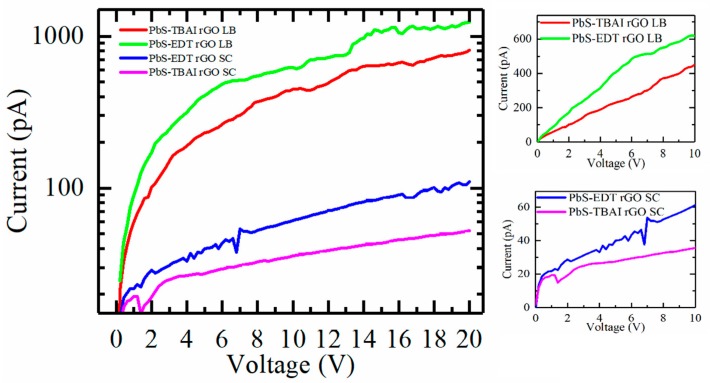
The I–V curves obtained for EDT and TBAI treated samples, prepared using LB and spin-coating techniques. Right panels show the linearity of the I–V curves.

**Table 1 nanomaterials-10-00723-t001:** The calculated specific conductance obtained for the LB and spin-coated samples.

Sample	Specific Conductance, S/m	Sample	Specific Conductance, S/m
LB (EDT)	4.3 × 10^–4^	Spin-coat. (EDT)	7.5 × 10^–5^
LB (TBAI)	2.7 × 10^–4^	Spin-coat. (TBAI)	7.2 × 10^–5^

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
