# Peer review of "Ligand-Assisted Formation of Graphene/Quantum Dot Monolayers with Improved Morphological and Electrical Properties"

_nanomaterials, 2020, doi:10.3390/nano10040723_

Round 1

Reviewer 1 Report

The article entitled "Ligand-assisted formation of graphene / quantum  dots monolayers with improved morphological and electrical properties" by the authors A. Litvin et al. is quiet interesting for the scientific community due that they use the Langmuir-Blodgett technique to fabricate thin films of PbS QDs,
attached to rGO by the MPTS linker. They clearly show that electrical properties of the films are much better that the ones done by drop casting. So I recommed this article  for publication 

Author Response

We are very thankful to the reviewer for the evaluation of our paper

Reviewer 2 Report

Litvin and colleagues describe a new method for the deposition of high quality film of PbS nanocrystals and graphene oxide. They analyzed the morphology, optical and electrical properties of the produced films, confirming the highest performance in comparison with films obtained by spin coating method.

I have only a few minor comments about the text:

  • Is the morphology similar throughout the entire film surface? Could the authors show more AFM or SEM images taken in different positions of the film to attest the high quality of the sample?
  • PL decay curves show a strong reduction of lifetime of PbS in presence of rGO. The time scale used in figure S3 is too large to measure that short lifetime. Does it measure with higher time resolution?
  • On line 96, a dot is missed after “under gentle stirring”. The dot after “12 h” has to be removed.    

I would suggest the publication of the manuscript in “Nanomaterials” journal after minor revisions.  

Author Response

We are very thankful to the reviewer for the evaluation of our paper and for the valuable comments which we have addressed in our revised manuscript. Our point by point responses to the individual comments is attached. All the answers are marked in red, while all the changes in the revised manuscript are marked in blue. 

Reviewer 3 Report

The authors investigate the “upscaling of PbS QD-rGO films by the Langmuir-Blodgett technique for optoelectronic devices” an interesting field as indicted in the introduction. However, some key points should be addressed before considering the manuscript. Although the authors claim a good performance of the resulting films, there is no figures of merit indicated such as responsivity that allows to compare with the literature. Here other issues to solve:

-how the authors decide the surface pressure selected for the deposition? Did they carry out preliminary experiments depositing films at different points or Brewster angle microscopy monitoring to confirm the coverage degree achieved? These experiments should be included in the SI.

-how reproducible are the Langmuir isotherms done by compression?

-there is no image demonstrating the coverage of the 3x2 cm2 area

-regarding the electrical tests of the films, why authors just show the I-V curves in the positive side? What patterning they used for the characterization? A comparable figure of merits such as responsivity should be provided to compare the systems and with the literature

-considering that the deposition is carried out on a water subphase, how the exposure with water affects to the final optoelectronic properties of the films, concretely to the PbS QDs? Control experiments even just using QDs should be done and included in the SI.

Author Response

(The authors gave the same response as above.)

Round 2

Reviewer 3 Report

Just minor spell check required.